# Dirichlet Process Prior for Student's t Graph Variational Autoencoders

Yuexuan Zhao [1,2] and Jing Huang [1,2,*]

1   College of Computer Science and Technology, Jilin University, Changchun 130012, China;
    zhaolx18@mails.jlu.edu.cn
2   Key Laboratory of Symbolic Computation and Knowledge Engineer, Ministry of Education, Jilin University,
    Changchun 130012, China
*   Correspondence: huangjing@jlu.edu.cn

**Abstract:** Graph variational auto-encoder (GVAE) is a model that combines neural networks and Bayes methods, capable of deeper exploring the influential latent features of graph reconstruction. However, several pieces of research based on GVAE employ a plain prior distribution for latent variables, for instance, standard normal distribution (N(0,1)). Although this kind of simple distribution has the advantage of convenient calculation, it will also make latent variables contain relatively little helpful information. The lack of adequate expression of nodes will inevitably affect the process of generating graphs, which will eventually lead to the discovery of only external relations and the neglect of some complex internal correlations. In this paper, we present a novel prior distribution for GVAE, called Dirichlet process (DP) construction for Student's t (St) distribution. The DP allows the latent variables to adapt their complexity during learning and then cooperates with heavy-tailed St distribution to approach sufficient node representation. Experimental results show that this method can achieve a relatively better performance against the baselines.

**Keywords:** neural network; Bayesian nonparametric; Graph variational auto-encoder; Student's t distribution; Dirichlet process; network representation learning



## 1. Introduction

Variational autoencoders (VAEs), graph variational autoencoders (GVAEs) and their variants [1–3], the deep generative models that have emerged in recent years, integrate the auto-encoder and variational inference, not only keep the excellent expression of the neural network but solve the randomness of probabilistic latent variables in the generation process. At present, the development of VAEs in the probability models has two primary directions. The first is to design a more expressive variational posterior, the other one is to replace the standard normal distribution prior of VAEs with learning more intricate prior [4–8]. Although works in the first direction have made some signs of progress in various kinds of learning tasks, delving into learning more about whether the variational posterior is valid is still interesting. Reference [9] proved that even with a very strong posterior, VAEs always can not match the learned posterior with N(0, I) prior. So we focus on learning a prior with abundant semantics for GVAEs.

The performance of GVAEs is mainly dependent on the validity of their data representation. Learning the representation of graph data, also known as network representation learning (NRL) [10–12], encodes nodes in the network and denotes them in a unified low-dimensional latent space. Hence, one way to improve GVAEs prior is to use more flexible probability distribution modeling latent variables. Gaussian distribution [13–16] is the most widely used probability distribution of latent variables in VAE models [17–19]. This kind of phenomenon is because of the well-known "central-limit theorem" [20]. If the sample size is large enough, the distribution of the samples' mean is approximately Gaussian, irrelevant to the distribution of samples in the population. The mean value of the samples'

mean distribution does not have to be zero, it depends on the value range of samples. More specifically, if there are a set of non-negative samples, their mean value is also non-negative. In addition, Gaussian distribution is the natural selection in the case of known mean and variance owing to its information entropy is the largest under this condition. However, it also has some disadvantages, especially its "empirical rule", which makes up about 99.7% of the numerical values distributed within three standard deviations centered on the mean. When the latent space dimension increases, its ability to capture more information will decline. Nevertheless, the choice of heavy-tailed Student's t (St) distribution [21–24] can significantly remedy this defect. For the reason that it can adaptively model the networks according to the degree of freedom (dof) parameter. The dof parameter controls the curve shape changes of the St distribution. The smaller the dof, the flatter the St distribution curve. Conversely, the larger the dof, the curve is closer to the Gaussian distribution with the same mean and scale parameters. Because most of the real-world networks are not random, the degree distribution in different networks more or less has the power-law property [25,26]. For instance, in the American airline network, nodes are airports, and the links between them are airlines. The nodes such as John F. Kennedy International Airport and Los Angeles International Airport are hubs with the power-law property. However, there are just several hubs in a network, and most nodes have a handful of links [27]. In addition, each latent vector is a node representation, so that we can fit the structure of real-world networks by applying the more general St distribution to model node representations. Concretely, ensuring the variational posterior of latent representation is close to a St distribution with zero mean and one scale, while dof remains unchanged [4]. They solve the problem that Gaussian prior never rejects outliers in the modeling. Furthermore, they deal with data with image characteristics, which is different from this paper.

As we probe into the data with St distribution, it is a pivotal point that how to statistically model data at an applicable complexity. If finite models used for analysis, we must specify the number of latent features in advance, but it is unknown beforehand and needs to learn from the data. In the paper citation network, latent features are represented by the keywords of a paper. If the latent dimension is small, the latent features are general keywords, such as data mining and the uncertainty of artificial intelligence. If the latent dimension is large, the latent features may be more specific keywords, such as anomaly detection, Bayesian network. By adding latent dimensions, the topic related to the paper can be dug out; then we can find neglected correlation relationship of some papers. However, if the latent dimension is too large, some useless latent features will appear, such as Chebyshev's inequality. Papers that apply the same inequality are regarded as relevant, but this decision is inaccurate in most cases.

The Bayesian nonparametric approach [28–31] gives us a way to solve this problem, which assumes that latent structures grow with data so that an infinite number of latent features can exist in the model. Under this condition, the determination of model complexity is incorporated into the data analysis process without setting manually based on expert knowledge or supervision information. The Dirichlet Process (DP) [32–34] is a typical Bayesian nonparametric method, which defines a binary matrix and each row of the matrix represents a node representation, each dimension captures a specific aspect of nodes. DP, as a prior of St distribution, can find possible features of all nodes in networks and also help discover important missing information or links.

In this paper, we propose DP-St prior for GVAE. In detail, we use St distribution as the probability distribution of a latent variable and solve the problem of limited detection ability of the traditional Gaussian distribution. At the same time, we apply the DP to explore more prior knowledge of nodes in a network, solve the problem that a fixed capacity can not precisely capture the complexity of the available networks well. We integrate DP and St distribution to construct an unbounded latent representation with sufficient expression ability, longer tails, and a deeper dimension to provide tremendous flexibility for latent variables. Our contributions are summarized as follows:

- We proposed DP-St prior, DP is the prior of St distribution and combined it with GVAE for network representation learning. The DP-St prior is a heavy-tailed nonparametric process, which can adapt to the dimension of latent variables, generate manifold prior knowledge for GVAE;
- We illustrate that the adjustment of the heavy-tailed level of St distribution is conducive to filter out useless latent features and retain the most representative ones. It can help detect missing links in the network and accurately model the real-world networks.
- In the experiments, the improvement of link prediction and semi-supervised node classification tasks are demonstrated. In particular, our model can effectively improve the prediction accuracy of link prediction in low-dimensional latent variables caused by St distribution modeling. It can also obtain the optimal prediction results in the semi-supervised node classification task when the labeled training data is small.

## 2. Related Work

Exploring the proper form of latent representation is an essential task for many representation learning applications. Moreover, we have seen that learning a prior of latent variables is indispensable to the VAE models. From the theoretical perspective, some recent studies demonstrate the necessity of learning VAE prior. Reference [35] highlighted the significance of the probability distribution of latent space by analyzing some variants of variational evidence lower bound objective (ELBO). Since it is necessary to complete the VAE optimization process by maximizing the ELBO, and this process is equivalent to minimizing the KL divergence between the variational posterior and prior of latent variables. Furthermore, Reference [36] illustrates the negative impact of applying a simple prior for VAE models. That is to say, it will result in few active latent variables, and finally produce over-regularized models with unpromising latent representation, which is not conducive to the tasks of presentation learning.

Considering the importance of latent representation for carrying out various learning tasks, some models improved VAE from the perspective of the Gaussian distribution of latent variables, which is commonly used in most VAE models and their variants. Reference [37] elaborate that applying the nonparametric process to a neural network can automatically ascertain some critical parameters in the model, such as model capacity. They use the stick-breaking process to generate latent variables, and the Stick-Breaking VAE model is constructed by various parameterization methods. This model shows more discriminative ability of latent representation than Gaussian VAEs. Reference [4] replace the Gaussian distribution of latent variables in the traditional VAE models with a higher error-tolerant St distribution and apply the standard St distribution as a prior, and demonstrate that heavy-tailed latent variables can aggrandize their flexibility when the data contains outliers. Nonetheless, only aggrandizing the tail thickness is far from enough to make the latent variables adapt to the dimension with sufficient flexibility and obtain ample prior knowledge to represent the data. Nonetheless, only enlarging the tail thickness is far from enough to make the latent variables adapt to the dimension with sufficient flexibility and obtain ample prior knowledge to represent the data. All the works mentioned above are used to deal with tasks related to image data, while we intend to do the presentation learning tasks on network data.

There are also a few works that employ the DP prior. In [38], they focus on the task of various classification through learning logistic-regression methods, and they illustrate that the DP is helpful to learn the similarity between different classification tasks. The DP-GP-LVM (Dirichlet Process-Gaussian Process-Latent Variable Models) [39] algorithm presents a Bayesian nonparametric latent variable model, apply the Gaussian process prior and DP prior at the same time. More precisely, they are used for producing latent variables and learning structure, respectively. However, neither of these works has considered the application of DP prior to latent variables. Our model takes full advantage of DP as a nonparametric method in the adaptive dimension and facilitates the learning of appropriate representation for latent variables.

## 3. Background

### 3.1. Dirichlet Process

In the network representation learning, it is pivotal to learn an appropriate representation for each node, which is demonstrated by a binary vector, every dimension of it indicates a latent feature the node may possess. The Bayesian nonparametric method is conducive to determining the number of latent features. The Dirichlet process (DP), as a widely used Bayesian nonparametric method, places a prior probability distribution over the sparse latent feature matrix, denoted $Z \sim DP(\alpha)$, whose number of rows is finite and columns are unbounded. The DP is defined as:

$$Z_n \mid G \sim G,$$
$$G \sim DP(\alpha, G_0), \tag{1}$$

where $n = 1, 2, ..., N$ for $N$ nodes in a network. The node representation $Z_n$ is drawn from a prior probability distribution G. G is sampled from the DP with a base distribution $G_0$ and a positive scale parameter $\alpha$.

In addition, the product of DP is a discrete distribution. Hence, we can apply the stick-breaking process to represent it more intuitively. The stick-breaking process is used to generate infinite-dimensional discrete variables. This procedure requires sampling a vector from a base distribution $Beta(1, \alpha)$ over $[0, 1]$ for each node, and all elements of the vector sum to one. Then, Reference [40] takes this process as the basis for the stick-breaking generation, and completes the DP with stick-breaking construction by arranging feature weights in descending order:

$$b_{(k)} \sim Beta(1, \beta),$$
$$\mu_{(k)} = b_{(k)} \prod_{l=1}^{k-1} (1 - b_{(l-1)}), \tag{2}$$

where $k = 1, 2, ..., K$, $\beta$ is the hyperparameter of Beta distribution, $b_{(k)}$ is the kth sample from the Beta distribution, $\mu_{(k)}$ is the kth weight from the DP process, and $\mu_{(1)} > \mu_{(2)} > ... > \mu_{(K)}$ indicates the descending ordering of $\mu_1, \mu_2, ..., \mu_K$.

### 3.2. Student's t Distribution

In the real world, many observed phenomena do not follow the frequently-used Gaussian distribution, and a more permissive distribution should be used to describe them. The Student's t distribution (St) [41], as the most frequently used heavy-tailed distribution, differs significantly from the Gaussian distribution, the thickness of tails is modulated by the dof parameter $v$. The smaller the dof parameter $v$, the flatter the St distribution, meaning the lower the middle of the curve, the higher the tails rise. In Figures 1 and 2, we compare the St distribution with other similar probability distributions. As the dof $v = 1$, the curve is the Cauchy distribution, and as the dof $v = \infty$, the curve is the Gaussian distribution. Overall, the lower dof $v$ in distribution will result in the more permissive of outliers. The probability density function (pdf) of the multivariate St distribution is:

$$f(x) = \frac{\Gamma(\frac{v+n}{2})}{\Gamma(\frac{v}{2})\sqrt{(\pi v)^n det(\Sigma)}} [1 + \frac{(x-\mu)^T \Sigma^{-1} (x-\mu)}{v}]^{\frac{v+n}{2}}, \tag{3}$$

where random variable $x \in \mathbb{R}^n$, with mean parameter $\mu \in \mathbb{R}^n$, scale matrix $\Sigma \in \mathbb{R}^{n \times n}$, dof parameter $v$, the diagonal elements of $\Sigma$ are $\sigma^2$, and $n$ is the dimension of $x$.

We can use the reparameterization trick to reparameterize two samples $u \sim N(0, I)$, $w \sim Gamma(\frac{u}{2}, \frac{1}{2})$ by the transformation $T(u, w) = u\sqrt{\frac{v}{w}}$, samples $T(u, w) \sim St(0, I, v)$ of the standard St distribution can be obtained from that [42], and then the linear transformation $t = \mu + \sigma T(u, w)$ gives us $t \sim St(\mu, \sigma, v)$.

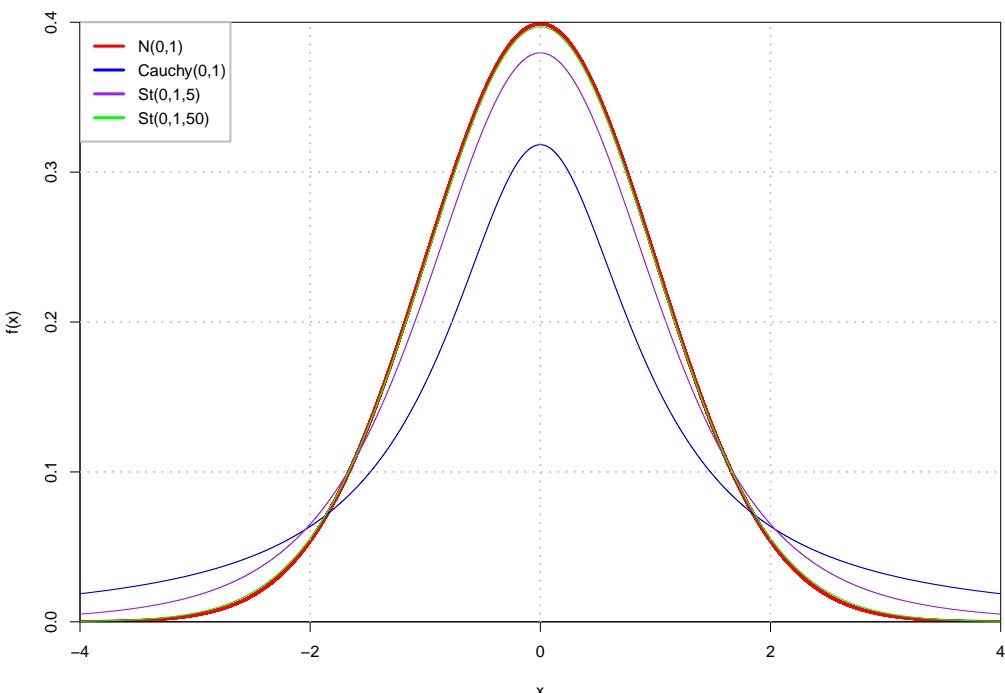

**Figure 1.** Several probability distribution function curves of common probability distribution. They share the same mean and scale parameters, and there are two St distribution with dof parameter 5 and 50 respectively.

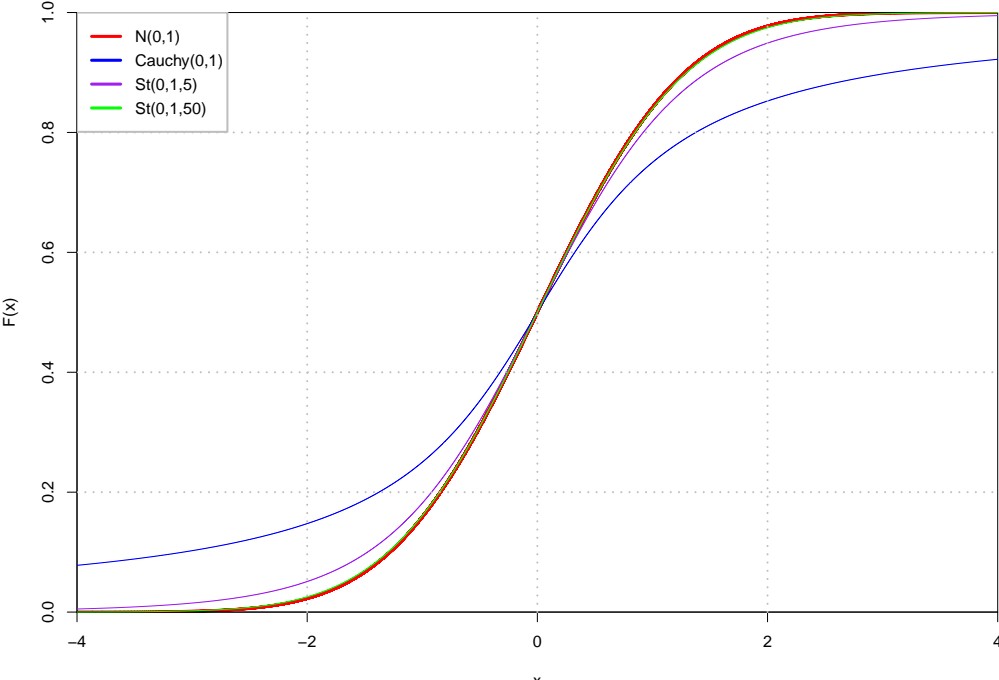

**Figure 2.** Several cumulative distribution function curves of common probability distribution. They share the same mean and scale parameters, and there are two St distribution with dof parameter 5 and 50, respectively.

## 4. Method

### 4.1. DP-St GVAE

In this section, we will introduce the heavy-tailed priors based on Bayesian nonparametric methods in detail, which is DP construction for St distribution priors. Utilizing the

augment of the prior distribution complexity, latent variables can be stimulated to learn more features, and more expressive latent representation can be obtained to reduce the reconstruction error.

Considering the St distribution as the prior distribution of latent variables, the bilateral tail is thicker, which can cover more latent features with higher probability, compared with the most commonly used Gaussian distribution. As the features increase, we have to allow higher dimensional latent representation to pertain to the nodes. Therefore, we hope to use DP to generate the mean parameter $\mu$ and scale parameter $\sigma$ of the St distribution. This method allows us to infer which factors have significant effects, and the dimensions of latent variables. For the last parameter, dof parameter $v > 0$ controls the tail shape of the pdf curve of St distribution, with the increase of dof $v$, the curve of the latent variables changes gradually from the heavy-tailed Cauchy distribution to the Gaussian distribution, which makes us find the curve that fits the observed data most in the procedure of learning the dof parameter.

In Figure 3, we show the framework of the DP-St GVAE model in detail. First, we assume that the observed data, adjacency matrix $\mathbf{A} \in \mathbb{R}^{\mathbf{N} \times \mathbf{N}}$ and attribute matrix $\mathbf{X} \in \mathbb{R}^{N \times C}$, are used as the inputs of encoder to generate the latent variables $\mathbf{Z} \in \mathbb{R}^{N \times K}$, where the $N$ observations come from a graph undirected attributed graph $\mathbf{G}(\mathbf{V}, \mathbf{E}, \mathbf{X})$. $\mathbf{V}$ is the node set $\{v_1, v_2, ..., v_N\}$, and $\mathbf{E}$ is a unweighted edge set, $e_{ij} = 1$ means there is a undirected link between node $v_i$ and $v_j$. We put a St distribution over the latent space, and place the outputs of GVAE's encoder over the mean, scale, and dof parameters of St distribution:

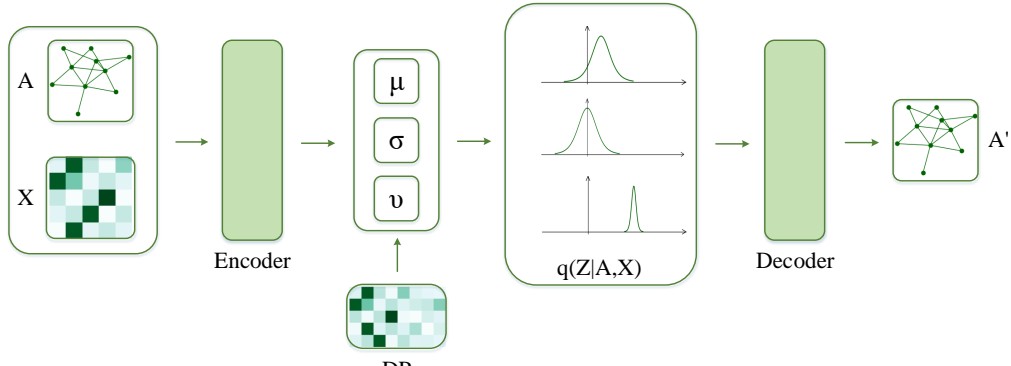

**Figure 3.** The architecture of the DP-St GVAE. The encoder is a graph convolutional neural network that integrates the network structure $A$ with node feature $X$ and, by extension, generates the node representation $Z$. The decoder reconstructs a graph from the inner product of the node representation $Z$.

$$q_\phi(\mathbf{Z} \mid \mathbf{A}, \mathbf{X}) = St(\mathbf{f}_{1,\phi}(\mathbf{A}, \mathbf{X}), \mathbf{f}_{2,\phi}(\mathbf{A}, \mathbf{X}), \mathbf{f}_{3,\phi}(\mathbf{A}, \mathbf{X})), \tag{4}$$

where $\mathbf{f}_{1,\phi}, \mathbf{f}_{2,\phi}, \mathbf{f}_{3,\phi}$ denote the neural networks with parameter $\phi$.

Second, we replace the standard Gaussian prior with a DP-St distribution. The DP is composed of a discrete base distribution $G_0$ and a concentration parameter. To explicitly illustrate the base distribution $G_0$, the stick-breaking process is a good choice. It generates unbounded sticks $b_{nk} \in [0, 1]$ independently from a beta distribution:

$$p(b_{nk} \mid \beta) = Beta(b_{nk} \mid 1, \beta). \tag{5}$$

From the procedure above, we obtain an infinite dimensional vector with components $\pi_k(\mathbf{b}) = b_k \prod_{l=1}^{k-1} (1 - b_{l-1})$, which is used as the initialization of mean and scale parameters in St prior of each node, and the dof parameter is initialed from a Uniform distribution:

$$p(\mathbf{Z}_n \mid \beta) = St(\mathbf{Z}_n \mid \pi_{1,n}(\beta), \pi_{2,n}(\beta), Uni(\alpha)). \tag{6}$$

In this way, we generate a latent representation of infinite dimensions for each node, and the DP allows latent variables to adapt complexity in the learning process. Moreover, it enables almost all nodes to seek out the most powerful features, reduced the error caused by the traditional allocation mode of weak discrimination. Finally, we make full use of the information inside the latent representation, and reconstruct a graph through the function $g(\cdot)$ with parameter $\theta$:

$$p_\theta(\mathbf{A} \mid \mathbf{Z}) = Bern(\mathbf{A} \mid g_\theta(\mathbf{Z})). \tag{7}$$

*4.2. Variational Inference for DP Prior*

We find that there is no direct method to deal with the posterior distribution of latent variables in the condition of DP-St prior. Variational inference opens up our train of thought and leads to a deterministic way for the approximate calculation of the likelihoods and posteriors. We denote the observed adjacency matrix $A$ and attribute $X$ as $W = \{A, X\}$, the hyperparameter *psi*, the log posterior distribution of latent variable $Z$ is:

$$logp(Z \mid W, \psi) = logp(W, Z \mid \psi) - logp(W \mid \psi). \tag{8}$$

However, we don't compute this posterior distribution directly due to the calculation of normalizing constant, and through some transformation, we can obtain the log marginal probability of the observation $W$:

$$logp(W \mid \psi) = log \int p(W, Z \mid \psi)dZ. \tag{9}$$

This equation is difficult to compute because of the dependency relationship between the latent variables and observed data.

In this work, we use the mean-field variational method, converting to an optimization problem from handling the posterior distribution problem. On this basis, the Kullback-Leibler (KL) divergence between posterior distribution $q_\phi(Z \mid W)$ and true posterior $p(Z)$ need to be optimized in training:

$$KL[q_\phi(Z \mid W) \parallel p(Z)] = \mathbb{E}_q[logq_\phi(Z \mid W)] - \mathbb{E}_q[logp(Z)] + logP(W). \tag{10}$$

The upper bound of KL divergence between variational posterior $q(z) = St(\mu_1, \Sigma_1, v_1)$ and St prior $p(z) = St(\mu_2, \Sigma_2, v_2)$ is [43]:

$$
\begin{aligned}
& KL[q(z) \parallel p(z)] \\
& \leq \frac{1}{2} \ln \frac{det\Sigma_2}{det\Sigma_1} + \frac{n}{2} \ln \frac{v_2}{v_1} + \ln \Gamma(\frac{v_1 + n}{2}) - \ln \Gamma(\frac{v_1}{2}) \\
& - \ln \Gamma(\frac{v_2 + n}{2}) + \ln \Gamma(\frac{v_2}{2}) - \frac{v_1 + n}{2}[\psi(\frac{v_1 + n}{2}) - \psi(\frac{v_1}{2})] \\
& + \frac{v_2 + n}{2} \ln(1 + \frac{1}{v_2}tr(\Sigma_2^{-1}\widetilde{\Sigma}_1) + \frac{1}{v_2}tr[\Sigma_2^{-1}(\mu_1 - \mu_2)(\mu_1 - \mu_2)^T]),
\end{aligned}
\tag{11}
$$

where $\psi(\cdot)$ is digamma function, $\widetilde{\Sigma}_1 = \frac{v_1}{v_1 - 2}\Sigma_1$.

Minimizing the Equation (10) can be translated into maximizing the lower bound of the log-likelihood $logp(W)$. We finally achieve the objective function:

$$L(\mu, \sigma, v, \phi) = \mathbb{E}_{q_\phi(Z \mid A, X)}[logp(A \mid Z)] - KL[q_\phi(Z \mid A, X) \parallel p_{\mu,\sigma,v}(Z)]. \tag{12}$$

## 5. Experiments

We report the performance of our DP-St GVAE on two tasks—unsupervised link prediction and semi-supervised node classification. Our model has been put into effect on three benchmark citation datasets (Cora, Citeseer, and PubMed) [44], and the datasets are

described in Table 1 in detail. Each node in these datasets represents literature, and has a unique attribute vector, the connection between literature indicates that there is a reference relationship between them.

**Table 1.** Details of the datasets.

|          | Cora | Citeseer | PubMed |
|----------|------|----------|--------|
| **Nodes**    | 2708 | 3327 | 19,717 |
| **Edges**    | 5429 | 4732 | 44,338 |
| **Features** | 1433 | 3703 | 500 |
| **Labels**   | 7    | 6    | 3 |

In all experiments, the models are realized by TensorFlow, and ADAM is used for training. The encoder is composed of GCN [45] with parameter $\phi$, and we let function $g(\mathbf{Z})$ use activation function tanh for the transformation $\mathbf{Z}\mathbf{Z}^T$. In our DP-St GVAE model, the sticking-breaking hyperparameter $\beta$ is set to 10, and the Uniform distribution hyperparameter $\alpha$ is set to 10.

*5.1. Link Prediction*

To assess our model's ability to perform link prediction tasks, we compared it with two algorithms: GVAE [3] and standard tGVAE [4]. Our model improves the problem that the standard St distribution as a prior can not fully discover the latent information for a large portion of nodes in networks, and in our experiments, it is applied to the network data, whose encoder and decoder are changed to use the same settings as our method, then we denoted it as St GVAE.

For this task, we will follow the settings of GVAE and make the encoder use GCN of two layers to conduct four groups of comparative experiments, the dimensions of latent space are 8, 16, 32, and 64, and the corresponding GCN hidden layer sizes are 16, 32, 64, and 128 respectively. All experiments use a learning rate of 0.01. When the learning rate is 0.1, it is difficult for the model to converge in training. When the learning rate is 0.001, the model converges very slowly, and its performance in learning tasks is lower than that in the learning rate of 0.01. The validation set and the test set contain 5% and 10% of the network edges. AUC (area under ROC curve) and AP (average accuracy) are used as the assessment criteria for the correct classification of edge types in the networks.

We show link prediction performance on three citation datasets in Tables 2–4, and our model is almost better than the contrastive approaches. As we can see from the experimental results, with the augment of the latent space dimension, the predictive performances of GVAE and St GVAE models are increasing, and our model presents a trend like a parabola. This is because our model can automatically select the most appropriate latent dimension, and the latent features under this dimension are conducive to the reconstruction of network structure. Moreover, there are St GVAE with a large predicted improvement span in the low-dimension interval, and GVAE with a uniform rising trend about the dimension. When the dimension of latent space is low, such as 8, the performance of St GVAE is the weakest among the three models, because, in the case of fewer latent features, the use of heavy-tailed St distribution can, to some extent, obscure the differences between various features. This indicates the importance of constructing the St distribution with DP because DP can help find the most critical features for each node among these few latent features. This ranking of importance can digest the disadvantage of heavy-tailed distribution in low-dimensional latent space.

**Table 2.** Link prediction performance comparison of different dimensions for Cora.

|  | 8 Dimensions | | 16 Dimensions | | 32 Dimensions | | 64 Dimensions | |
|---|---|---|---|---|---|---|---|---|
|  | **AUC** | **AP** | **AUC** | **AP** | **AUC** | **AP** | **AUC** | **AP** |
| GVAE | 91.0 | 91.1 | 91.4 | 92.6 | 92.7 | 93.6 | 93.3 | 94.4 |
| St GVAE | 85.5 | 83.4 | 91.0 | 89.9 | 92.1 | 89.7 | 95.0 | 95.2 |
| DP-St GVAE | 92.3 | 93.9 | 94.1 | 95.1 | 94.4 | 95.5 | 93.5 | 94.8 |

**Table 3.** Link prediction performance comparison of different dimensions for Citeseer.

|  | 8 Dimensions | | 16 Dimensions | | 32 Dimensions | | 64 Dimensions | |
|---|---|---|---|---|---|---|---|---|
|  | **AUC** | **AP** | **AUC** | **AP** | **AUC** | **AP** | **AUC** | **AP** |
| GVAE | 89.5 | 90.5 | 90.8 | 92.0 | 91.0 | 92.1 | 91.1 | 92.5 |
| St GVAE | 80.6 | 80.0 | 90.9 | 89.9 | 95.5 | 95.0 | 95.5 | 96.1 |
| DP-St GVAE | 95.5 | 96.2 | 96.9 | 97.2 | 97.3 | 97.6 | 96.9 | 97.2 |

**Table 4.** Link prediction performance comparison of different dimensions for PubMed.

|  | 8 Dimensions | | 16 Dimensions | | 32 Dimensions | | 64 Dimensions | |
|---|---|---|---|---|---|---|---|---|
|  | **AUC** | **AP** | **AUC** | **AP** | **AUC** | **AP** | **AUC** | **AP** |
| GVAE | 92.7 | 92.9 | 94.4 | 94.7 | 95.6 | 95.6 | 96.5 | 96.5 |
| St GVAE | 89.9 | 87.8 | 97.4 | 97.3 | 97.8 | 97.5 | 97.8 | 97.6 |
| DP-St GVAE | 96.4 | 97.1 | 97.1 | 97.0 | 97.3 | 97.4 | 97.9 | 97.8 |

Now, to understand the latent representation $\mathbf{Z}$, we visualize its posterior probability, $q_{\phi}(\mathbf{Z})$, for GVAE and DP-St GVAE models on Cora and Citeseer datasets. With regard to the node selection for visualization, we first randomly select ten nodes with the most and the least number of latent features with high probability respectively, and then randomly choose 10 nodes in the remaining node-set. In addition, the dimensions of latent variables in the DP-St GVAE model are automatically determined, and for the convenience of comparison, we select similar dimensions for the GVAE model. From Figures 4a and 5a, we can see that many nodes have multiple latent features with high probability, and the latent representation contains rich information and complex correlation. However, in Figures 4b and 5b, the latent representation has higher sparsity, and the latent features of each node are more explicit. In the same dimension of latent space, the latent representation generated by GVAE will bring more disturbing information and severely restrict the nodes to find the most representative latent features. After adding a DP-St prior, the various information can be screened, and those useless features that have a negative impact on the target task can be discarded to produce appropriate latent representation.

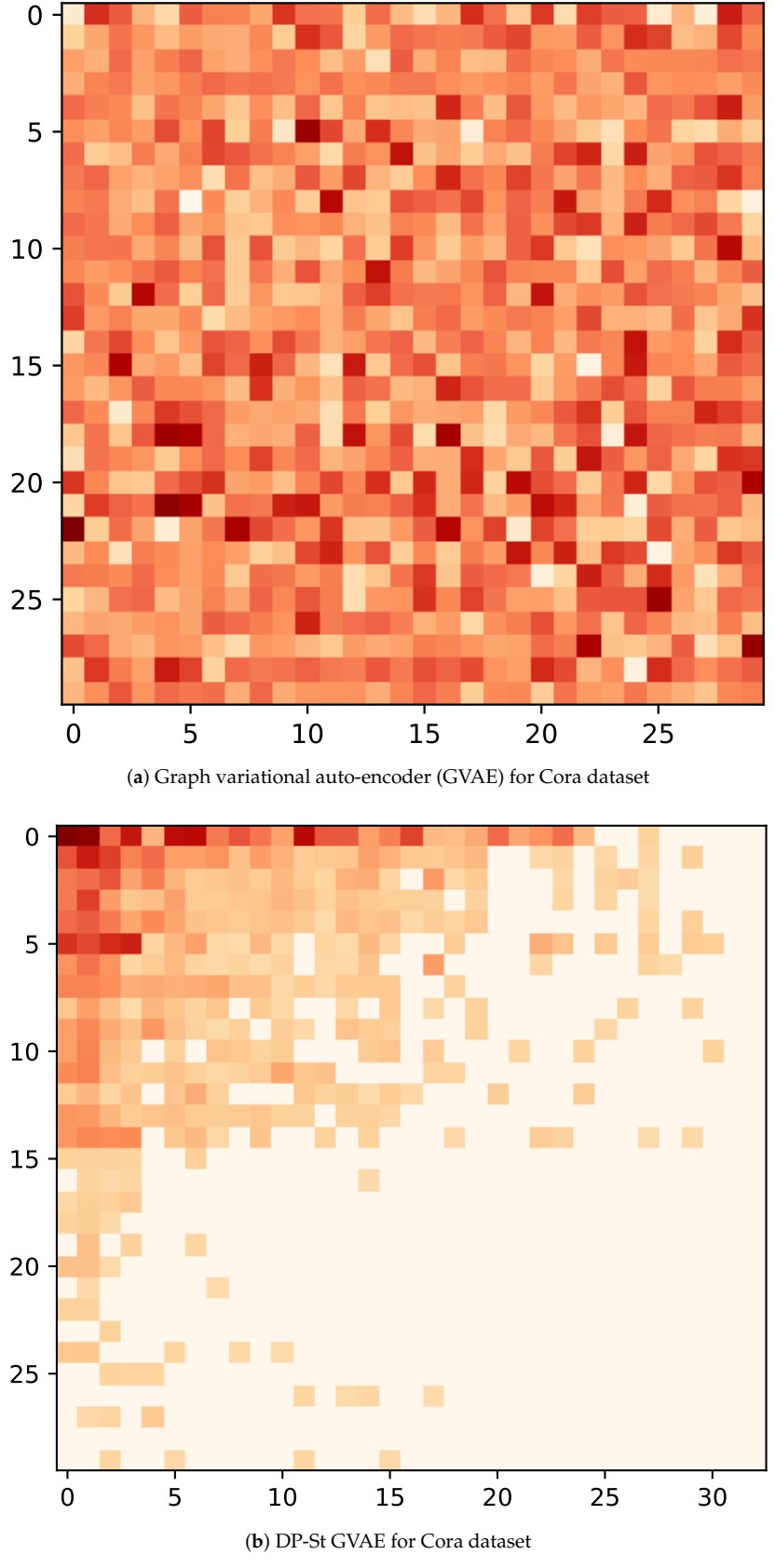

(**a**) Graph variational auto-encoder (GVAE) for Cora dataset

(**b**) DP-St GVAE for Cora dataset

**Figure 4.** Latent representation visualization for Cora dataset. The horizontal axis is the dimension of the hidden variable, and the vertical axis is the number of nodes. (**a**) the latent variable learned by GVAE. (**b**) the latent variable learned by DP-St GVAE. The darker the higher probability nodes possess the latent features.

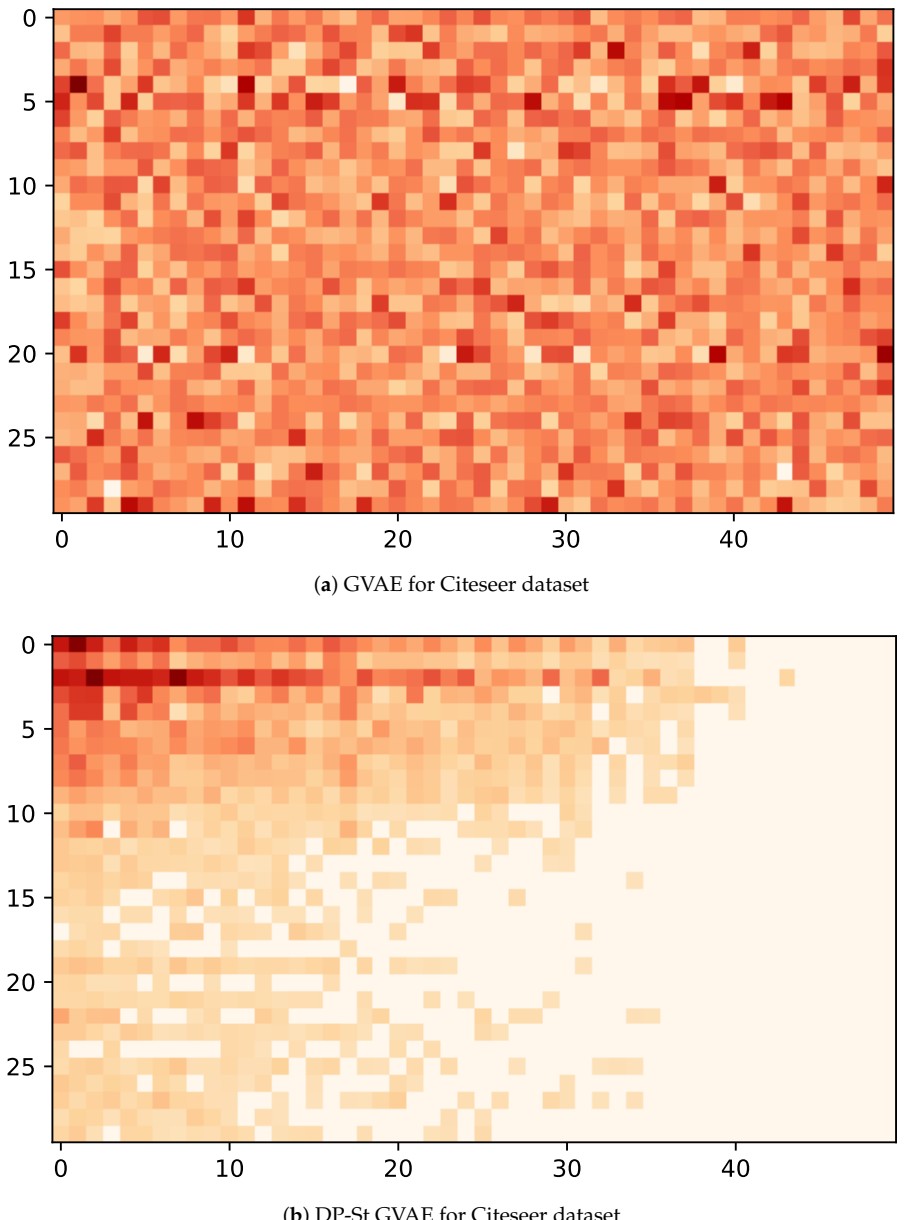

(**a**) GVAE for Citeseer dataset

(**b**) DP-St GVAE for Citeseer dataset

**Figure 5.** Latent representation visualization for Citeseer dataset. The horizontal axis is the dimension of the hidden variable, and the vertical axis is the number of nodes. (**a**) the latent variable learned by GVAE. (**b**) the latent variable learned by DP-St GVAE. The darker the higher probability nodes possess the latent features.

### 5.2. Semi-Supervised Node Classification

For the semi-supervised node classification task, the goal is to make the label prediction for nodes without labels. So, in this task, we add the cross-entropy term of labels to the objective function (12). We make use of the random split partition method [46], randomly dividing all nodes in each dataset into three subsets, training set, validation set, and test set. The number of known labels has been separated into three groups, with 5, 10, and 20 labels per class. For this experiment, we compare our model with the GVAE and St GVAE models used in the link prediction task and added several related works in this task—Graph attention network (GAT) [47], and Graph convolutional neural network (GCN) [45].

Tables 5–7, respectively, show the results on the Cora, Citeseer, and PubMed datasets. The experimental results illustrate that our model achieves the optimal classification result on the PubMed dataset, because there is heavy-tailed degree distribution, and our

model can precisely grasp the hidden information and obtain a more obvious accuracy improvement. On the Cora and Citesser datasets, our model is able to achieve more competitive results when the number of labels is known to be smaller. When the number of known labels is tiny, we can see that the node classification performance of GVAE is less than 50% on the Citeseer dataset, suggesting a high error rate in the generated node representation. Furthermore, it is obvious that the St GVAE model improves the prediction accuracy by replacing the Gaussian distribution with the St distribution, indicating that the traditional Gaussian distribution leaves out some essential information for nodes, and the St distribution can avoid this problem. Finally, when we construct a DP prior for the St distribution, each node can capture the most important hidden information, which provides convenience for discovering the correlation between nodes, and thus the affiliation of nodes.

**Table 5.** Semi-supervised node classification prediction accuracy for Cora.

|  | 5 Labels | 10 Labels | 20 Labels |
|---|---|---|---|
| GAT | $70.4 \pm 3.7$ | $76.6 \pm 2.8$ | $79.9 \pm 1.8$ |
| GCN | $70.0 \pm 3.7$ | $76.0 \pm 2.2$ | $79.8 \pm 1.8$ |
| GVAE | $68.4 \pm 2.4$ | $70.3 \pm 2.7$ | $71.6 \pm 1.4$ |
| St GVAE | $69.8 \pm 2.4$ | $71.0 \pm 1.5$ | $72.2 \pm 2.0$ |
| DP-St GVAE | $75.1 \pm 2.6$ | $76.9 \pm 2.7$ | $79.3 \pm 2.4$ |

**Table 6.** Semi-supervised node classification prediction accuracy for Citeseer.

|  | 5 Labels | 10 Labels | 20 Labels |
|---|---|---|---|
| GAT | $56.7 \pm 5.1$ | $64.1 \pm 3.3$ | $67.6 \pm 2.3$ |
| GCN | $58.5 \pm 4.7$ | $65.4 \pm 2.6$ | $67.8 \pm 2.3$ |
| GVAE | $49.1 \pm 1.8$ | $52.1 \pm 1.4$ | $52.4 \pm 2.5$ |
| St GVAE | $52.5 \pm 1.3$ | $53.1 \pm 2.9$ | $53.8 \pm 1.5$ |
| DP-St GVAE | $59.2 \pm 2.5$ | $64.5 \pm 2.5$ | $67.9 \pm 1.8$ |

**Table 7.** Semi-supervised node classification prediction accuracy for PubMed.

|  | 5 Labels | 10 Labels | 20 Labels |
|---|---|---|---|
| GAT | $68.0 \pm 4.8$ | $72.6 \pm 3.6$ | $76.4 \pm 3.0$ |
| GCN | $69.7 \pm 4.5$ | $73.9 \pm 3.4$ | $77.5 \pm 2.5$ |
| GVAE | $74.6 \pm 1.9$ | $74.9 \pm 2.3$ | $75.4 \pm 3.0$ |
| St GVAE | $76.2 \pm 2.6$ | $77.5 \pm 3.3$ | $79.0 \pm 3.9$ |
| DP-St GVAE | $76.4 \pm 3.9$ | $78.4 \pm 1.9$ | $79.8 \pm 1.2$ |

## 6. Conclusions

In this work, we propose a DP construction for Student's t distribution prior, and combine it with GVAE. We solve the inaccuracy problem of modeling real-world networks by manually setting latent dimensions through the adaptive complexity of Bayesian nonparametric methods. In addition, the advantages and disadvantages of applying Gaussian distribution in traditional VAEs are analyzed at length, and we have illustrated that the application of Student's t distribution can effectively improve the limitation of effective latent features. Here, our DP-St GVAE is used to the network structure data, and the combination of the infinite-dimensional nonparametric generation process and heavy-tailed distribution plays an important role in network presentation learning tasks. In the future, our work will proceed, and there are two primary directions: (i) applying the DP-St to deep neural networks so that the problem of future research will break the limitation of graph structure data; (ii) using other Bayesian nonparametric methods for further improving the prediction accuracy of network representation learning tasks.

**Author Contributions:** Conceptualization, Y.Z.; methodology, Y.Z.; software, Y.Z.; validation, Y.Z. and J.H.; formal analysis, Y.Z. and J.H.; investigation, Y.Z. and J.H.; resources, Y.Z. and J.H.; data curation, Y.Z.; writing—original draft preparation, Y.Z.; writing—review and editing, Y.Z. and J.H.; visualization, Y.Z.; supervision, J.H.; project administration, J.H.; funding acquisition, J.H. All authors have read and agreed to the published version of the manuscript.

**Funding:** This research was funded by the National Natural Science Foundation of China under grant number 61572226, 61876069; Jilin Province Key Scientific and Technological Research and Development project under grant numbers 20180201067GX, 20180201044GX; Jilin Province Natural Science Foundation under grant No. 20200201036JC.

**Data Availability Statement:** Data is contained within the article. The data presented in this study are available in [Revisiting Semi-Supervised Learning with Graph Embeddings. Zhilin Yang, William W. Cohen, Ruslan Salakhutdinov. ICML 2016.], the computer code is available at [https://github.com/kimiyoung/planetoid (accessed on 9 February 2021)].

**Conflicts of Interest:** The authors declare no conflict of interest.

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
