# Peer review of "Dirichlet Process Prior for Student’s t Graph Variational Autoencoders"

_futureinternet, doi:10.3390/fi13030075_

Round 1
Reviewer 1 Report
- Line 35 - I recommend removing the question from the text and reformulating the whole sentence. According to me, questions do not look too professional in expert papers.
- Line 37 - Could you clarify whether the mean value is always zero? This should be mentioned in the paper.
- Line 38 - I recommend providing a brief explanation of the central limit theorem more extensively.
- Line 45 - Please explain DOF more extensively.
- Line 61 - It is stated: "If it is too small, the heavy-tailed property will be limited and the latent variables with St distribution will bear a striking resemblance to that using Gaussian, on the contrary, the excessive amounts of latent features will generate plenty of useless dimensions for the node representation."Could you specify more precisely what too small and the excessive amounts represent as both of them are very relative terms? Maybe, some real-world examples could be added.
- Line 66 - you mentioned that "an infinite number of latent features can exist in the model" In the paper, could you clarify whether the fact, that an infinite number of latent features can exist, complicates the implementation into real-world systems? Is it possible to specify memory requirements for such systems?
- Line 68 - in the paper, could you explain the term early assignment more extensively?
- In Fig. 1, I recommend adding labels of both axes.
- Fig. 1, are also distributions with non-zero mean common?
- Fig. 1, I think it would be helpful to show also the cumulative distribution functions.
- Line 138 - could you specify whether there is a range of values that alfa can take?
- Line 142 - It is stated: "In the real world, many observed phenomena do not follow the frequently-used Gaussian distribution" In the paper, could you discuss how to effectively find out that the observed phenomenon does not follow Gaussian distribution?
- Line 180 - could you describe the considered graphs more extensively? Are they weighted, do they contain multiple edges, etc?
- Line 190 - could you specify the parameters of the uniform distribution?
- In tables 2-4, you use maximally 64 dimensions. Is there any limit for the number of dimensions? Must it be always the power of two?
- Line 224 - it is stated that "All experiments use learning rate 0.01". In the paper, could you provide the information how different values of this parameter can affect the achieved results?
- Line 285 - This sentence is hard to understand and needs revision. "In this work, we propose a DP construction for Student’s t distribution prior, and combine it with GVAE, makes the progress of generating node representation with sufficient latent information by means of adaptive latent dimension"
- English contains several minor mistakes, for example:
- "s" is missing in the third person singular
- incorrectly used plurals- missing commas
- missing spaces - incorrectly used spaces
- I do not recommend starting sentences with And
- I recommend substituting don’t with do not, which is more formal
etc.
Author Response
we would like to thank you for your comments and suggestions. We have carefully analysed each of them to make sure that they are adequately incorporated and addressed in our revised version as attached. In the submitted document follows, we provide our detailed responses to the reviewers’ comments and at the same time indicate the corresponding changes made.

Reviewer 2 Report
This article presented a novel prior distribution for GVAE, called Dirichlet process construction for Student’s t distribution. This is rather a theoretical work, and the novelty is to show that the proposed distribution improves the accuracy of link prediction. Generally, the structure of this paper is correct, in line with the standards of writing scientific texts.
The disadvantage of the work is a small scientific contribution. The proposed distribution is a combination of well-known methods. The equations are poorly commented and not all variables appearing in the formulas are described. There are no punctuation marks between the equations. There is also a large number of typos in the text. It makes it difficult to read the work.
Author Response
For the disadvantage of small scientific contribution, we rewrote the innovation points, .As suggested by the Reviewer, we added comments on some variables of Eqs. (2), i.e. β is the hyperparameter of Beta distribution, b(k) is the kth sample from the Beta distribution, μ(k) is the kth weight from the DP process. We added punctuation marks between the equations, i.e., Eqs. (4)-(10). We scrupulously checked the spelling and made corrections in the revised paper.
Round 2
Reviewer 1 Report
The authors addressed most of my comments. I did not find these two answers (worth adding to the paper) in the paper:
- Answer 5 - I did not find this answer in the paper. Could you add it to your paper?
- Answer 6 - would it be possible to show also the cumulative distribution functions? I think that it would be helpful for readers to see them.
Author Response
We added the explanation of "early assignment" in answer 5, and showed the cumulative distribution function figure in our revised paper.
Reviewer 2 Report
I read the article again and noticed that the authors improved the work. Now, the article can be taken into consideration for publication on the Future Internet.
Author Response
Thank you for your comments.